# Development of New Edible Biodegradable Films Containing Camu-Camu and Agro-Industry Residue

**DOI:** 10.3390/polym16131826

**Published:** 2024-06-27

**Authors:** Huéberton Barbosa Naves, Ana Paula Stafussa, Grasiele Scaramal Madrona, Fabrício Cerizza Tanaka, Fauze Ahmad Aouada, Márcia Regina de Moura

**Affiliations:** 1Programa de Pós-Graduação em Ciência dos Materiais, Faculdade de Engenharia do Campus de Ilha Solteira—SP, Universidade Estadual Paulista, Avenida Brasil 56, Ilha Solteira 15385-000, SP, Brazil; huebertonbarbosa@hotmail.com; 2Programa de Pós-Graduação em Ciência de Alimentos, Universidade Estadual de Maringá, Avenida Colombo 5790, Maringá 87020-900, PR, Brazil; anastafussa@gmail.com; 3Departamento de Engenharia de Alimentos, Universidade Estadual de Maringá, Avenida Colombo 5790, Maringá 87020-900, PR, Brazil; gsmadrona@uem.br; 4Programa de Pós-Graduação em Engenharia de Alimentos, Faculdade de Zootecnia e Engenharia de Alimentos, Universidade de São Paulo, Campus Fernando Costa, Avenida Duque de Caxias Norte, 225, Pirassununga 13635-900, SP, Brazil; tanaka.fabricio@usp.br; 5Departamento de Física e Química, Faculdade de Engenharia do Campus de Ilha Solteira—SP, Universidade Estadual Paulista, Avenida Brasil 56, Ilha Solteira 15385-000, SP, Brazil; fauze.aouada@unesp.br

**Keywords:** *Myrciaria dubia* (H. B. K.) Mcvaugh, gelatin, glycerol, biodegradable, antioxidants

## Abstract

The use of edible films has garnered significant interest in the food and environmental sectors due to their potential to prevent food deterioration and their biodegradability. This study aimed to develop and characterize edible films based on camu-camu residue, gelatin, and glycerol, evaluating their solubility, thermal, degradability, antioxidant, and water vapor permeability properties of the gelatin matrix. This is the first study incorporating camu-camu into a gelatin and glycerol matrix. The films produced with camu-camu residue were manageable and soluble, with some non-soluble residues, providing a shiny and well-presented appearance. In the biodegradation results, samples 3 and 4 appeared to degrade the most, being two of the three most affected samples in the triplicate. The films showed degradation modifications from the third day of the experiment. In the germination and plant growth analysis, sample 4 exhibited satisfactory development compared to the other samples, emerging as the sample with the best overall result in the analyses, attributed to a 13.84 cm increase in the growth of the upper part of the seedling. These results indicate that the produced materials have potential for food packaging applications.

## 1. Introduction

In recent decades, the food industry has been characterized by the large volume of waste it produces. Approximately one-third of the food produced for human consumption, around 1.3 billion tons per year, is lost or wasted worldwide. The most significant loss is observed in fruits and vegetables, accounting for approximately 0.5 billion tons. In developing countries, a substantial portion of this loss occurs during processing. However, significant losses also occur during post-harvest and distribution stages [1,2].

In Brazil, food processing activities have expanded significantly to meet the market demand for processed foods [3]. The processing of fruits and vegetables generates a substantial amount of waste, including peels, seeds, pulp, and bagasse. These by-products are rich sources of nutrients, particularly biopolymers (such as polysaccharides and dietary fiber) and bioactive compounds [4]. In this context, it is important to find applications for camu-camu residue, especially considering the fruit’s high vitamin C content.

Biopolymers, such as polysaccharides and proteins derived from agricultural by-products, have been proposed for the formulation of biodegradable materials. These biopolymers are susceptible to biodegradation (except when subjected to severe chemical modifications), originate from renewable sources, and are non-toxic to the soil and environment [5]. Due to these environmental concerns, significant attention has been directed towards the development of edible and biodegradable films. Edible films are packaging materials made from non-toxic macromolecules designed to control barriers such as moisture permeability and other factors that accelerate food degradation, including methylene, oxygen, and light exposure. Additionally, these films can incorporate additives such as antimicrobials and antioxidants, aimed at expanding the range of potential applications. These films can also enhance nutritional value and improve sensory characteristics when consumed along with the product to which they are applied. Moreover, they can serve as an alternative to petroleum-derived materials, reinforcing the sustainability of the process [6,7].

Films derived from vegetables exhibit low to moderate oxygen permeability and acceptable mechanical properties. For instance, films incorporated with kale pulp are suggested as an alternative for the production of edible packaging and coatings [8]. However, there are still limited studies on the use of fruit and vegetable processing residues in the development of biodegradable films [9,10]. In studies conducted by Rangaraj, Rambabu, Banat, and Mittal [11], the effect of incorporating extracts derived from date syrup waste into gelatin films for food preservation was examined. It was reported that the extract releases antioxidant molecules in a slow and controlled manner within the films, thereby preventing the formation of peroxides and consequent degradation reactions, such as lipid auto-oxidation. Another study by Santos and collaborators [12] reported antioxidant activities in gelatin films incorporated with Malpighia emarginata waste extract. The addition of 4% residue was shown to reduce the formation of malonaldehyde and total carbonyl compounds. Consequently, the resulting packaging exhibits promising properties for preserving meat products, potentially extending their shelf life.

In an effort to find a beneficial application for camu-camu residue, considering the fruit’s richness in vitamins, the residue was incorporated into a gelatin and glycerin matrix, and its characteristics were analyzed. The fruit is primarily sold on a small scale at local markets in the production region, with the majority being sold as frozen pulp. Although relatively unknown domestically, it is highly sought after by Japanese, American, and European markets, and is exported in refrigerated containers [13]. It is important to highlight that this is the first study on gelatin and glycerin films incorporated with camu-camu residues. Previous studies integrating camu-camu residues into polymeric film matrices noted that the bioactive compounds of camu-camu could impart antioxidant properties to the packaging. This observation was confirmed by Ju and Song [14], who developed biodegradable films based on starch and camu-camu with these specific characteristics.

In terms of food packaging in Brazil, according to the Brazilian Packaging Association (ABRE), plastics account for 37.47% of the total value of packaging production. This material consequently represents a significant volume of improper disposal, in addition to the emission of toxic and polluting gasses during production stages. However, due to their diversity and versatility, petroleum-derived polymers provide technological advancements, energy savings, and several other benefits to society through the production of a variety of products. Bioplastics, such as edible films developed from biopolymers like hydroxypropyl methylcellulose (HPMC), chitosan, alginates, and gelatins, are non-toxic natural alternatives that are extremely promising for replacing conventional packaging. Some of these biopolymers are also abundant sources of antimicrobial action [9,10,15].

One of the main characteristics of gelatin is its high hygroscopic nature. When in contact with cold water, its particles swell, increasing their mass by approximately ten times. When heated above its denaturation temperature (40 °C), gelatin forms a colloidal solution with water and gels upon cooling. This behavior makes films made from gelatin more likely to dissolve when immersed in water above this temperature [16,17].

Given the current scenario of escalating waste generation from fruit and vegetable processing and their potential in the synthesis of new products, coupled with the environmental impact of plastic packaging widely used in the food industry, this study aimed to evaluate the potential of using fruit waste in the production of biodegradable films. Specifically, it explored their potential application in managing and protecting solid foods with high water activity and significant levels of phenolic compounds and lipids. Additionally, by utilizing an Amazonian fruit, this study aims to highlight the potential of Amazonian raw materials, demonstrating how their comprehensive exploitation can address environmental challenges and stimulate the local circular economy.

## 2. Materials and Methods

### 2.1. Material

Gelatin powder P.A. (bovine) (CAS [9000-70-8], Dinâmica, Indaiatuba—SP); Bi-distilled Glycerol U.S.P. (P.M. 92.09), Synth, Diadema—SP; camu-camu residue dried at 50 °C for 24 h and ground at the State University of Maringá-PR. The residue was obtained from the unused fruit parts left over from pulp production for ice cream. It is rich in fibers and may contain a considerable amount of vitamin C and antioxidants.

### 2.2. Preparation of the Gelatin Filmogenic Solution

A subjective analysis and a pH analysis were conducted to select the best film, focusing on characterizing positive results. Gelatin solutions at 2% (*w*.*v*^−1^) (the film matrix) were produced by hydrating 1 g of powdered gelatin in 49 g of distilled water for one hour. After hydration, the sample was conditioned in a water bath, monitoring the temperature up to 50 °C for complete dissolution. Finally, the solution was stirred on a magnetic stirrer (500 rpm) until it cooled down to 30 °C. Bi-distilled glycerin was used at concentrations of 25% and 35% (*w*.*w*^−1^) relative to the mass of the polymer. The amount of glycerin was chosen based on previous studies conducted by the research group. This compound was added to improve the manageability of the samples.

### 2.3. Preparation and Weighing of Camu-Camu Extract

The camu-camu extract was dried at 50 °C for 24 h. Subsequently, the extract was manually macerated in a porcelain crucible to obtain fine grains, maximizing homogenization in the matrix. Two concentrations were utilized for the characterizations (0.25 g and 0.5 g).

### 2.4. Incorporation of the Extract into the Matrix

The films were produced using the casting technique, incorporating glycerin and camu-camu extract. Casting involves the deposition of the filmogenic solution onto a polyester support on a leveled bench, followed by drying at room temperature. This method is widely used by the research group for obtaining this type of packaging [18].

The experimental plan consisted of four samples: sample 1 contained 0.25 g of extract and 0.25 g of glycerin incorporated into the matrix; sample 2 contained 0.5 g of extract and 0.25 g of glycerin; sample 3 contained 0.25 g of extract and 0.35 g of glycerin; and sample 4 contained 0.5 g of extract and 0.35 g of glycerin. All samples were subjected to stirring using a Mod.752 magnetic stirrer (Fisatom São Paulo/SP, Brazil) for two hours at a speed of 500 rpm and also homogenized with an Ultra-Turrax homogenizer (Marconi-MA102, Piracicaba/SP, Brazil) for 30 min at a speed of 12,000 rpm.

Additionally, results will include samples of pure gelatin with 0.25 g of glycerin (referred to as “pure 25%”) and pure gelatin with 0.35 g of glycerin (“pure 35%”).

### 2.5. Determination of Thickness

The thickness of the films was measured using a digital micrometer (No. 7326, Mitutoyo Manufacturing, Kawasaki, Japan). Five analyses were conducted in different areas of each film.

### 2.6. Scanning Electron Microscopy (SEM)

The morphology analyses of the films were conducted using a computerized scanning electron microscope (ZEISS model EVO LS15, Oberkochen, Germany) with a voltage of 12 kV. The films were affixed to double-sided carbon tape and metalized using a Sputter Coater (Quorum, model Q150 T, Lewes, UK) for 1.5 min to deposit a thin layer of gold on them.

For this analysis, the samples were placed in a glass desiccator with silica for 24 h to remove moisture present in the film. Only after this period were the films cut into square shapes, each with a length of 5 cm, for analysis [19].

### 2.7. Quantification of Total Phenolic Compounds and Antioxidant Capacity

Samples 3 and 4 and the pure 35% were analyzed immediately after preparation (1:3 *w/v* with methanol) for extraction. The extracts were obtained by homogenization using centrifugation (15 min, 4000 rpm) and filtration (using filter paper) [20].

Total phenolic compounds were determined using the Folin–Ciocalteu colorimetric method [21] with minor modifications. The results were expressed in milligrams of gallic acid equivalent per 100 g of sample on a dry basis (mg EAG g^−1^ dry basis).

The free radical scavenging potential was determined by the 2,2-diphenyl-1-picrylhydrazyl (DPPH) assay according to Thaipong et al. [22]. The free radical scavenging activity by the 2,2′-azino-bis(3-ethylbenzothiazoline-6-sulfonic acid) (ABTS) radical was determined according to Re et al. [23]. The FRAP assay was performed according to the methodology of Benzie and Strain [24]. All antioxidant activities were expressed in micromoles of Trolox equivalent per gram of dry weight (µmol TE g^−1^ dry weight).

### 2.8. Water Vapor Permeability (WVP)

For the WVP analyses, the method adapted from ASTM (1980) was used [25], as described by McHugh and collaborator [26].

Plates with the films were placed in an oven with a controlled temperature (25 ± 2 °C) and humidity (50 ± 3%). Silica was used to control internal humidity. Plates were weighed hourly to determine the WVP. To calculate the WVP, Equations (1)–(3) were used.
(1)$$Water\;Vapor\;Transmission\;Rate\;(WVTR)=\frac{film\;mass\;loss}{film\;area}$$
(2)$$\mathrm{WVTR}=\frac{m_{w}\times P\times D\times ln\left[\frac{\left(p-p_{2}\right)}{\left(p-p_{1}\right)}\right]}{R\times T\times z}$$
(3)$$WVP=\frac{WVTR}{\left(p_{2}-p_{3}\right)}\times y$$
where *m_w_* is the molar mass of water (18 g·mol^−1^), *D* is the diffusivity of water vapor through air at 298 K (0.102 m^2^·s^−1^), *P* is the total pressure (1 atm), *p*_1_ is the vapor pressure at 298 K, *R* is the gas constant (82.1 × 10^−6^ m^3^·atm/g·mol·K), *z* is the average height the inert gas reaches, *p*_2_ is the partial vapor pressure at the bottom of the film, *p*_3_ is the partial vapor pressure at the top of the film, and *y* is the average film thickness.

### 2.9. Soil Biodegradation

To evaluate the soil biodegradation of the developed films, a commercial plant substrate from the brand Carolina Soil was used. Its composition includes peat, vermiculite, class A agro-industrial organic residue (roasted rice husk), and limestone, as described on the label.

Following the methodology applied by Pavoni et al. [27], film samples of 25 mm^2^ were arranged in triplicate on a plastic mesh and buried in the substrate at a depth of approximately 20 mm in a polypropylene (PP) container with an internal diameter of 14 cm. The biodegradation test was conducted at room temperature, and deionized water was sprayed daily to maintain humidity. Commercial films of low-density polyethylene (LDPE) and cellulose were included as controls.

As reported by Abe et al. [28], measuring biodegradation by weight loss of bioplastics is extremely challenging. Therefore, the samples were carefully removed from the substrate after 3, 5, 10, 15, and 30 days to evaluate the disintegration process through visual inspection.

### 2.10. Morphological Analysis

For the visual analysis, photomicrographs were taken using a Zeiss Optical Stereo Microscope (model Stemi 508, Oberkochen, Germany), in addition to photographs taken with a Nikon camera (model Coolpix P510, Tokyo, Japan). The morphological analysis was conducted using a scanning electron microscope (JEOL, model JSM-6510LV, Tokyo, Japan). Sample preparation involved drying at 60 °C in an oven without air circulation for 1 h. Subsequently, the samples were affixed onto a copper support with carbon tape and subjected to vacuum metallization with gold for 3 min using a Denton Vacuum (model Desk V, Moorestown, NJ, USA). All samples were stored in a desiccator before analysis.

### 2.11. Seed Germination in the Presence of Degraded Film

The methodology developed by Harada and collaborators [29] was adapted to evaluate the toxicity of the substrate for plant growth after the degradation of the films over 30 days. Black common bean (*Phaseolus vulgaris* L.) seeds were planted (7 seeds per container), using the plant substrate that was not exposed to the samples during the biodegradation experiment as a control. The seeds were watered with deionized water daily for 10 days. Subsequently, they were cleaned, and parameters including germination (%), growth (cm), and total biomass (g) were evaluated.

## 3. Results and Discussion

### 3.1. Gelatin Films (2%) Incorporated with Glycerin and Camu-Camu Residue

A plan was implemented for the production of films that, once ready, underwent a subjective evaluation to classify them based on their continuity (absence of fracture or rupture after drying), homogeneity (absence of particles visible to the naked eye, opacity), and handling (ability to handle the film without risk of breakage) [30].

In the subjective analysis, the films with the best continuity, homogeneity, and handling were those produced with a 2% (*w*/*v*) gelatin matrix, classified as samples 3 and 4. The films titled as samples 1 and 2 came in second place. Based on this analysis, the films named as samples 1 and 2 were eliminated from the preparation process due to their poor handling, roughness, and brittleness. In all samples prepared for analysis, the insoluble residue was incorporated.

### 3.2. pH Analysis of Gelatin Films (2%) Incorporated with Glycerin and Camu-Camu Residue

In addition to the pH analysis (Table 1), a viscosity analysis was conducted to assess the influence of the residue on the film matrix and its acidification.

Upon examining the results in the table, it can be observed that the presence of camu-camu residue slightly acidifies the filmogenic solution. This outcome is expected, as astringent fruits typically possess this characteristic, which aids in the formation of films by increasing solution viscosity. The existing literature also supports the notion that viscosity plays a crucial role in film formation [30].

### 3.3. Film Thickness

The average thickness of the films, measured with a precision of 0.001 mm in five different areas, ranged between 0.020 and 0.080 mm. This result is consistent with findings reported in other studies [31].

It is worth noting that since these films are incorporated with residues that are not entirely dissolved, the thickness varies depending on the presence of insoluble residues. This indicates that the surface exhibits heterogeneity in terms of thickness.

### 3.4. Scanning Electron Microscopy (SEM)

Based on the surface morphology of the films presented in Figure 1, as well as the cryogenic fractures of the cross-section of the 35% pure matrix films shown in Figure 1d, and sample films 3 and 4 depicted in Figure 1d and 1e, respectively.

The images in Figure 1d reveal a smooth and homogeneous surface, without phase separation or the formation of agglomerates. The images in Figure 1d,e, illustrating the surfaces of the polymer matrices and cryogenic fractures, exhibit surfaces with some agglomerates, indicating high similarity across all samples, considering the regions containing non-solubilized residue.

The alteration in the texture of both the surface and interior of the film with the addition of the camu-camu residue is likely attributed to the formation of agglomerates resulting from the interaction between the components of the gelatin film and the residue.

It is evident that there was a change in surface roughness and the formation of agglomerates upon the addition of the camu-camu residue, stemming from the approximation of polar chains within the polymeric matrix. However, as depicted in Figure 1e,f, the incorporation of camu-camu residue led to alterations in the microstructure of the films. Pores are noticeable in the films, consistent with findings by de Andrade [32], who investigated the incorporation of vegetable flour in edible films. This phenomenon arises from the integration of a hydrophobic component, rich in fiber, into a hydrophilic phase. Presumably, the addition of camu-camu residue caused some disruption in the interactions within the polymeric network, hindering chain alignment and resulting in system heterogeneity, as evidenced in the cross-sectional micrographs.

### 3.5. Quantification of Antioxidant Capacity

Table 2 presents the analysis of antioxidant capacity using DPPH, FRAP, and ABTS methods, expressed in μg Trolox/g of sample, and total phenolic compounds in mg EAG/g of sample.

Considering the values presented, it can be inferred that a very satisfactory result was obtained in the analysis of the pure residue. When compared with the films incorporated with residue, even at lower concentrations, the presence of antioxidants and phenolic compounds can be affirmed. This justifies the absence of detection in the film without residue. Sample 4 exhibited 56% more total phenolic compounds and, on average, 60% more antioxidant content than sample 3.

### 3.6. Water Vapor Permeability (WVP)

The results obtained for the water vapor permeability of the produced films are presented in Table 3. Water vapor permeability (WVP) occurs due to a difference in relative humidity (RH) between the cells used. This analysis measures the ability of water vapor to diffuse from a region with higher water vapor pressure to one with lower water vapor pressure [33].

It can be observed that the film produced solely from 2% gelatin with 25% and 35% glycerin exhibits significantly lower water vapor permeability values compared to the other two formulations of 2% gelatin with 35% glycerin incorporated with 0.25 g and 0.5 g of camu-camu.

As shown in Table 3, the WVP value increased from 1.146 to 2.72 g-mm/kPa-h-m^2^ when the residue was incorporated with 0.25 g of camu-camu, and from 1.146 to 2.712 g-mm/kPa-h-m^2^ when the residue was incorporated with 0.5 g of camu-camu.

This increase can be attributed to the presence of glucose found in fruits and in greater quantities in these two formulations, which impairs the barrier properties of the material. However, further analyses need to be performed to confirm this hypothesis [1,2].

Films of the same concentration, 2% gelatin with 35% glycerin, incorporated with 0.25 g of camu-camu, were compared using the same production process. When placed on the polyester base for drying, the insoluble substrate was not included; it was left at the bottom of the beaker.

It is observed that the film without residue had a WVP difference of approximately 20%, indicating that the incorporation of the insoluble substrate in the matrix during the drying process facilitates water vapor passage. This confirms the difference in WVP between the pure gelatin matrix and samples incorporated with camu-camu substrate, as analyzed in Table 3.

These results may be advantageous depending on their applications and are consistent with findings reported by Altiok et al. [34] and Kavoosi et al. [35]. For instance, if the application involves a food product that can tolerate this level of WVP.

### 3.7. Biodegradation of Samples

Figure 2 depicts samples before and after biodegradation over various time intervals. Visual analysis based on photographs revealed increased opacity, rigidity, and brittleness of the samples compared to their initial states. Changes in the films were evident from the third day of the experiment. However, by the end of the observation period, complete degradation had not occurred, possibly due to the presence of a plastic mesh support, which limited full contact between the sample’s bottom surface and the substrate.

Comparing the degradation of different materials over 30 days, it is evident that samples 3 and 4 exhibited the most significant degradation, with two out of the three triplicate samples being notably affected.

Additional photographs illustrating the degradation of sample 4 after 15 and 30 days, included in the Appendix A of this study, provide a more detailed view of the sample’s mass loss and darkening. This darkening likely resulted from hydrolysis induced by enzymes and microorganisms present in the degradation environment, owing to the nutrient-rich and fiber-rich composition of the material. These alterations may have been facilitated by substrate particles adhering to the surface, as well as the structural degradation of the material due to exposure to plant substrate, environmental microorganisms, and moisture [4].

### 3.8. Seeds Germinated with the Presence of Degraded Films

Figure 3 illustrates the results of germination under various conditions, demonstrating the significant potential of degraded films as substrates for planting.

Despite the observed reductions in aerial part growth, root growth, and total biomass in Figure 3b–d, the germination percentage depicted in Figure 3a indicates that in the presence of products resulting from the degradation of Gelatine 35% samples, Gelatine Agmag sample 3, and Gelatine Agmag sample 4, all seeds germinated. This percentage surpasses that of the control with pure substrate and sample 4, which exhibited a germination rate of 85.71%. This increase in germination, along with the enhanced growth of the aerial part, root, and total biomass compared to seeds with cellulose and LDPE, underscores the beneficial effects of the degraded film on seedling development.

Notably, sample 4 exhibited satisfactory plant growth compared to other samples, with the highest recorded growth in the upper part of the seedlings (13.84 cm). These findings suggest that the films developed in this study can indeed promote plant growth.

Seed germination after 10 days demonstrated significant development, indicating the positive contribution of the residue to seed growth.

Analysis of the graphs reveals that the presence of camu-camu residue significantly enhances seed germination and promotes seedling growth. Based on the results discussed in Section 3.7 and Section 3.8, it can be concluded that this new material is not only biodegradable but also that the products resulting from its degradation are unlikely to have a negative impact on the environment. On the contrary, they could potentially serve as compostable material, unlike conventional petroleum-based packaging.

## 4. Conclusions

Gelatin and glycerin films were obtained by incorporating dried camu-camu residues at concentrations of 0.25 g and 0.50 g. The films produced with the incorporation of camu-camu residue proved to be manageable and visually homogeneous, with the presence of some non-solubilized residues, yet exhibiting a bright and well-presented appearance.

The gelatin and glycerin films incorporated with camu-camu residues, with a mass of 0.5 g, exhibited the best interaction between the particles and the polymeric matrix due to the presence of fibers in the waste.

Through the experimental planning and subjective analysis used in the preparation of the films, it was possible to produce films with good homogeneity, continuity, and handling.

It was observed that the film without insoluble residue had a water vapor permeability (WVP) approximately 20% lower than the film with insoluble residue, indicating that the incorporation of the insoluble residue in the matrix collaborates with the passage of water vapor. This difference in WVP between the pure gelatin matrix (35%) and samples 3 and 4, which were incorporated with camu-camu residue, was significant for the study because, for future applications, the presence of non-solubilized residues in the film causes slight separation in the polymeric molecule.

In the scanning electron microscopy (SEM) analysis, it was identified that there was a change in surface roughness and the formation of agglomerates with the immersion of the camu-camu residue. Due to the approximation of the polar chains occurring in the polymeric matrix, the immersion of the camu-camu residue caused changes in the microstructure of the films.

The findings presented in this article suggest that films made of gelatin and camu-camu offer a cost-effective option with antioxidant properties that can inhibit certain reactions known to reduce the shelf life of food. This preservation of organoleptic characteristics ensures the quality of the product.

## Figures and Tables

**Figure 1 polymers-16-01826-f001:**
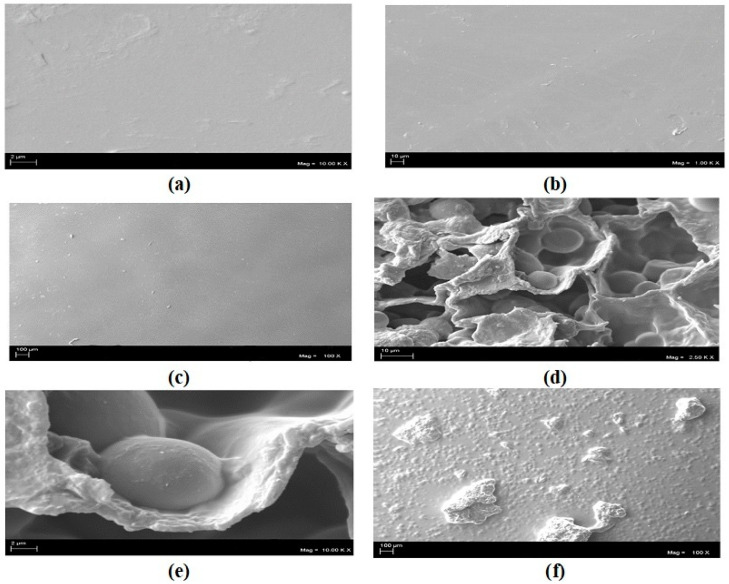
Surface micrographs of the 35% pure sample at nominal magnifications of 50.00× (**a**), 10.00× (**b**), and 100 K× (**c**). Fracture micrographs of the sample X at nominal magnifications of 50.00× (**d**), 10.00× (**e**), and 100 K× (**f**).

**Figure 2 polymers-16-01826-f002:**
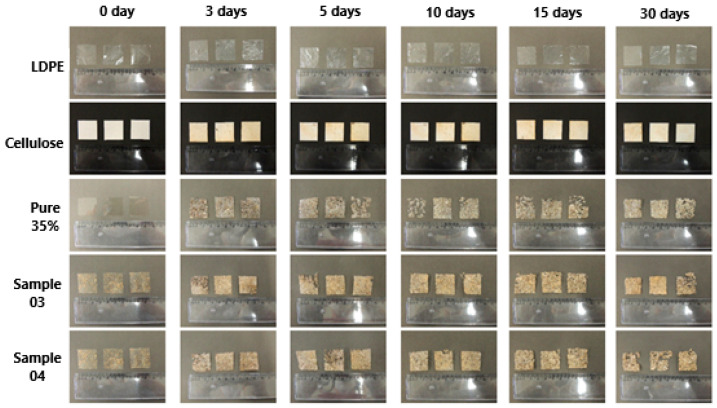
Photographs of samples before biodegradation and after periods of 3, 5, 10, 15, and 30 days.

**Figure 3 polymers-16-01826-f003:**
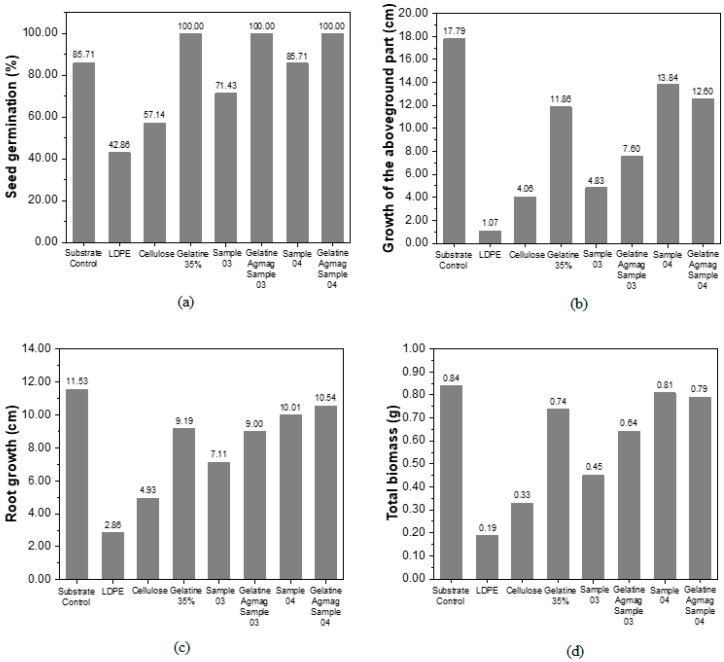
(**a**) seed germination (%), (**b**) growth of the aboveground part (cm), (**c**) root growth (cm) and (**d**) total biomass (g).

**Table 1 polymers-16-01826-t001:** Samples with different concentrations and their respective pH values.

Sample	Gelatin Concentration	Residue Concentration	Glycerin Concentration	pH
25% PURE	2% Gelatin matrix	---	0.25 g Glycerin	pH 5.24
35% PURE	2% Gelatin matrix	---	0.35 g Glycerin	pH 5.26
Sample 1	2% Gelatin matrix	0.25 g residue	0.25 g Glycerin	pH 4.50
Sample 2	2% Gelatin matrix	0.5 g residue	0.25 g Glycerin	pH 4.14
Sample 3	2% Gelatin matrix	0.25 g residue	0.35 g Glycerin	pH 4.50
Sample 4	2% Gelatin matrix	0.5 g residue	0.35 g Glycerin	pH 4.13

**Table 2 polymers-16-01826-t002:** Results of analysis of antioxidant capacity.

Samples	DPPH (mg Trolox/g Film)	FRAP (mg Trolox/g Film)	ABTS (mg Trolox/g Film)	Total Phenolic Compounds (mg EAG/g Film)
Sample 3	2.44 ^b^ ± 0.03	6.27 ^b^ ± 0.10	0.51 ^b^ ± 0.04	0.23 ^b^ ± 0.04
Sample 4	2.63 ^a^ ± 0.05	15.45 ^a^ ± 0.09	1.42 ^a^ ± 0.02	0.53 ^a^ ± 0.05
35% Pure	N.D. *	0.37 ^c^ ± 0.03	N.D. *	N.D. *
Substrate	35.07 ± 0.10	58.75 ± 0.02	49.79 ± 0.05	14.28 ± 0.08

Mean values ± standard deviation (SD); n = 3 (triplicate analysis); different superscript lowercase letters in the same column are significantly different (*p* ≤ 0.05). N.D. *—not detected.

**Table 3 polymers-16-01826-t003:** Standard deviation for analyzed samples.

Samples	WVP (g-mm/kPa-h-m^2^)	Deviation %
25% Pure	1.711	0.1082
35% Pure	1.146	0.0966
Sample 3	2.720	0.0900
Sample 4	2.712	0.0886

## Data Availability

The original contributions presented in the study are included in the article, further inquiries can be directed to the corresponding author.

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
