# Peer review of "Development of New Edible Biodegradable Films Containing Camu-Camu and Agro-Industry Residue"

_polymers, 2024, doi:10.3390/polym16131826_

Round 1
Reviewer 1 Report
Comments and Suggestions for Authors
Author Response
All points were considered and we thank you for the review. I am thankful for your time and consideration and do look forward to hearing from you, hopefully with a positive feedback.

Reviewer 2 Report
Comments and Suggestions for Authors
The article polymers-2986574 is devoted to the disposal of camu-camu waste. For the first time, it was proposed to use camu camu waste to produce biodegradable films based on gelatin with the addition of glycerin. The production meets the concept of a circular economy. The material in the article is well presented, accessible, understandable, and the article is well written. At the same time, they have critical comments, clarifications and questions.
Criticisms:
1) Gelatin is a polymer, but this is not new. The article is based on the introduction of camu camu into gelatin. Therefore, it is necessary to justify what polymers are contained in camu-camu so that this article can be published in the journal Polymers.
2) Please provide the complete chemical composition of the camu camu residue that you use and indicate what is the residue after? After the juice has been separated? Or is it a residue after extraction of biologically active substances? Then what solvent was used to extract?
3) Are gelatin films currently used for food products in the food industry? If yes, then please provide references to the literature. It’s not research, but a product on the market.
4) The annotation says that on the third day it is clear that the films have undergone biodegradation, how then can they prevent food spoilage? Will they be spoiled by food or spoiled along with food?
5) In the conclusions it is written “Incorporation of camu camu residues into the gelatin film caused a slight decrease in tensile strength and an increase in elongation,” but this data is not found either in the materials and methods, or in the experimental part. You need to add this information.
6) The introduction of camu camu into gelatin leads to a decrease in the strength of the films and accelerates biodegradation. Then the conclusion about the prospects of such films is not scientifically justified.
Small notes:
7) In the introduction, add the information, what are the reserves of camu camu in Brazil?
8) Lines 75-79. It is logical to end this paragraph with a conclusion about which products gelatin-based packaging is suitable for and which it is not.
9) Lines 96-97. Please justify why glycerol concentrations of 25 and 35% were chosen. Complete the article with information about why it is necessary to add glycerin to the film composition.
10) Line 99. Add data, with what and how was the extract macerated?
11) Line 106. How many grams of matrix are contained in samples 01-04%? Or what is the mass of samples 01-04? 1 gram?
12) Lines 300-302. Why do the authors conclude that camu camu is good for plants? The results are lower than in the control.
Reviewer's personal question:
Since the correspondent is heading a school on biodegradable films, please explain what the concept of edible film is? Is it assumed that the consumer will eat some product, for example, cheese, and eat the film in which the cheese is packaged?
Author Response

(The authors gave the same response as above.)

Reviewer 3 Report
Comments and Suggestions for Authors
Visual inspection is not an adequate technique to evaluate and to make a statement about biodegradeability of the samples. Visual inscpection can make a clarification of the results obtained by some other technique suitable for determination of biodegradaeabilty (FTIR, DP determination, gas measurement, etc.). The method you took to evaluate biodegradeabilty can only be a measure of decomposure, it does not mean that samples have biodegraded. in addition, how did you manage to maintain the soil humidity and stop water evaporation? didi you measure soil water capacity / retention / humidity?
why didn't you make the SEM microscopy of samples after soil burial?
line 283-284: how can you make a claim about the preseance of funghi without making any valid characterization?
Refrence list is not prepared according to Journal standards.
The germination test does not say a lot about the possible overall soil toxicity.
Comments on the Quality of English Languageminor gramatical errors
Author Response

(The authors gave the same response as above.)

Round 2
Reviewer 2 Report
Comments and Suggestions for Authors
The authors provided comprehensive answers to all questions. The article may be published as presented.
Author Response
Responses to reviewers
A careful review of the English was carried out. Thanks for the corrections
Reviewer 3 Report
Comments and Suggestions for Authors
Authors did not make impovement of the paper or did not gave a satisfactory clarification of the questionable parts of the manuscript.
minor grammar and style errors
Author Response

(The authors gave the same response as above.)
